
# Sensitivity to grid resolution in the ability of a chemical transport model to simulate observed oxidant chemistry under high-isoprene conditions

Karen Yu[1], Daniel J. Jacob[1,2], Jenny A. Fisher[3], Patrick S. Kim[2], Eloise A. Marais[1], Christopher C. Miller[1], Katherine R. Travis[1], Lei Zhu[1], Robert M. Yantosca[1], Melissa P. Sulprizio[1], Ron C. Cohen[4], Jack E. Dibb[5], Alan Fried[6], Tomas Mikoviny[7], Thomas B. Ryerson[8], Paul O. Wennberg[9,10], and Armin Wisthaler[7,11]

[1]School of Engineering and Applied Sciences, Harvard University, Cambridge, MA, USA
[2]Department of Earth and Planetary Sciences, Harvard University, Cambridge, MA, USA
[3]School of Chemistry, University of Wollongong, Wollongong, NSW, Australia
[4]Department of Chemistry, University of California, Berkeley, CA, USA
[5]Earth System Reserach Center, University of New Hampshire, Durham, NH, USA
[6]Institute for Arctic and Apline Research, University of Colorado, Boulder, CO, USA
[7]Department of Chemistry, University of Oslo, Oslo, Norway
[8]Earth System Research Laboratory, National Oceanic and Atmospheric Administration, Boulder, CO, USA
[9]Division of Geological and Planetary Sciences, California Institute of Technology, Pasadena, CA, USA
[10]Division of Engineering and Applied Sciences, California Institute of Tecnology, Pasadena, CA, USA
[11]Institute for Ion Physics and Applied Physics, University of Innsburck, Innsbruck, Austria

*Correspondence to:* Karen Yu (kyu@seas.harvard.edu)

**Abstract.** Formation of ozone and organic aerosol in continental atmospheres depends on whether isoprene emitted by vegetation is oxidized by the high-$NO_x$ pathway (where peroxy radicals react with NO) or by low-$NO_x$ pathways (where peroxy radicals react by alternate channels, mostly with $HO_2$). We used mixed layer observations from the SEAC$^4$RS aircraft campaign over the Southeast US to test the ability of the GEOS-Chem chemical transport model at different grid resolutions

5  ($0.25° \times 0.3125°$, $2° \times 2.5°$, $4° \times 5°$) to simulate this chemistry under high-isoprene, variable-$NO_x$ conditions. Observations of isoprene and $NO_x$ over the Southeast US show a negative correlation, reflecting in part the spatial segregation of emissions; this negative correlation is captured in the model at $0.25° \times 0.3125°$ resolution but not at coarser resolutions. As a result, less isoprene oxidation takes place by the high-$NO_x$ pathway in the model at $0.25° \times 0.3125°$ resolution (54%) than at coarser resolution (59%). The cumulative probability distribution functions (CDFs) of $NO_x$, isoprene, and ozone concentrations show

10  little difference across model resolutions and good agreement with observations, while formaldehyde is overestimated at coarse resolution because excessive isoprene oxidation takes place by the high-$NO_x$ pathway (which has high formaldehyde yield). Correlations of simulated vs. observed concentrations do not improve with grid resolution because finer modes of variability are intrinsically more difficult to capture. Higher model resolution leads to decreased conversion of $NO_x$ to organic nitrates and increased conversion to nitric acid, with total reactive nitrogen oxides ($NO_y$) changing little across model resolutions. In

15  the lower free troposphere, model output is similarly insensitive to grid resolution, indicating that the effect on export of ozone



and $NO_x$ is small. The overall low sensitivity of modeled concentrations to grid resolution implies that coarse resolution is adequate when modeling regional boundary layer chemistry for global applications.

## 1 Introduction

Global simulations of tropospheric chemistry present a major computational challenge. Chemical mechanisms typically include
over one hundred coupled species, with lifetimes ranging from less than a second to more than a year, interacting with transport on all scales from concentrated emission plumes to the remote troposphere. This complexity has hindered the inclusion of detailed tropospheric chemistry in climate models (NRC, 2012) and raises concern over spatial resolution. Global models typically have a horizontal resolution of $\sim$100 km, which does not properly resolve chemical gradients and may lead to large errors from non-linear chemistry and coupling to transport. Increasing resolution is computationally expensive and may require
trade-offs in other aspects of the model. Here we apply a global chemical transport model (GEOS-Chem CTM) to simulate extensive boundary layer observations of ozone and related species over the Southeast US from the SEAC[4]RS aircraft campaign (Toon et al., 2015). We explore how varying grid resolution from $4° \times 5°$ ($\approx$ 400x400 km$^2$) to $0.25° \times 0.3125°$ ($\approx$ 28x28 km$^2$) affects model results and the ability to simulate observations.

Ozone ($O_3$) production is central to driving the complexity of chemical mechanisms. It is controlled by reaction chains
involving hydrogen oxide radicals ($HO_x \equiv OH$ + peroxy radicals), nitrogen oxide radicals ($NO_x \equiv NO + NO_2$), halogen radicals, volatile organic compounds (VOCs), and a large ensemble of reservoir and product species. NO and VOCs are emitted by a wide range of sources, both natural and anthropogenic. $NO_x$ typically has a lifetime of hours while VOCs have lifetimes ranging from minutes to years (Atkinson and Arey, 2003). Their interactions result in a diversity of chemical regimes.

A number of studies have examined the effect of model resolution on ozone production in urban regions (Sillman et al., 1990;
Jang et al., 1995; Gillani and Pleim, 1996; Kumar and Russell, 1996; Chock et al., 2002; Esler et al., 2004). A typical result is that dilution from grid averaging causes positive bias in the ozone production efficiency (OPE) per unit $NO_x$ emitted (Liu et al., 1987). Liang and Jacobson (2000) showed that premature mixing of urban and background airmasses can lead to either overestimates or underestimates of OPE. Model simulations of ship plumes indicate particularly large ozone overestimates when mixing $NO_x$ from the plumes into otherwise clean grid cells (Davis et al., 2001; Vinken et al., 2011). On a global scale,
Wild and Prather (2006) found from an asymptotic error convergence method that grid averaging in a $2.8° \times 2.8°$ model caused a +4% bias in the global tropospheric ozone burden, with larger errors on regional scales. Ito et al. (2009) compared global models of varying resolutions and concluded that artificial mixing of biogenic VOC emissions into coarse grid cells yield excessive conversion of $NO_x$ to organic nitrate reservoirs, leading to release of $NO_x$ downwind under higher-OPE conditions and thereby causing excessive ozone production.

The interaction between $NO_2$ and OH can also lead to biases in model estimates of $NO_x$. Valin et al. (2011) found that large point sources of $NO_x$ suppress OH, producing long-lived $NO_x$ plumes. Dilution of these plumes into coarse resolution grid cells shortens the $NO_x$ lifetime, leading to underestimation of $NO_2$ columns over large sources and overestimation over small



sources. Yamaji et al. (2014) similarly found that coarse resolution CTMs tend to underestimate retrievals of $NO_2$ columns over industrial regions.

The SEAC[4]RS observations provide an opportunity to investigate the effect of chemical non-linearity in an environment with very high biogenic VOC emissions (mainly isoprene) and variable levels of $NO_x$ (mostly from combustion). The interactions between these species impact oxidant chemistry. Isoprene is oxidized by the OH radical on a time scale of an hour to produce isoprene peroxy radicals ($ISOPO_2$). These $ISOPO_2$ radicals may either react with NO to produce ozone (high-$NO_x$ pathway) or react by other channels (low-$NO_x$ pathways). Grid averaging may affect the ability to resolve the different pathways with implications for ozone, oxidant chemistry in general, and the formation of secondary organic aerosol (Marais et al., 2015).

We also use the SEAC[4]RS data to examine the effect of grid resolution on model ability to simulate variability in chemical concentrations, especially events at the high tails of the probability distributions that may be of particular interest. Higher resolution enables better simulation of chemical gradients, with some benefit for capturing extreme values (Zhang et al., 2011). However, Rastigejev et al. (2010) suggested that much of that advantage may be lost during long-range transport because of numerical diffusion in divergent flow. Higher model resolution does not always improve the correlations with observed concentrations (Kiley et al., 2003; Mathur et al., 2005; Arunachalam et al., 2006; Valari and Menut, 2008) because finer modes of variability are harder to capture than coarser modes (Fiore et al., 2003).

## 2 Methods

The GEOS-Chem simulation for the SEAC[4]RS period (August-September 2013) uses GEOS-5 data produced by GMAO at $0.25° \times 0.3125°$ horizontal resolution with 72 vertical layers (lumped to 47 layers in GEOS-Chem) and 3-h temporal resolution (1-h for surface quantities and mixing depths). Here we use that native resolution as reference for comparison to the $2° \times 2.5°$ and $4° \times 5°$ resolutions that would be used in global applications. Other GEOS-Chem studies of the SEAC[4]RS period apply the simulation with $0.25° \times 0.3125°$ resolution to interpret observations of organic nitrates (Fisher et al., 2015), ozone (Travis et al., 2015), aerosols (Kim et al., 2015), and formaldehyde (Zhu et al., 2015). Marais et al. (2015) uses $2° \times 2.5°$ resolution to simulate isoprene secondary organic aerosol in SEAC[4]RS. These references present relevant model descriptions and evaluations.

The $0.25° \times 0.3125°$ simulation extends over a continental-scale North America window ($130° - 60°$W, $9.75° - 60°$N) for two months (August-September 2013). Initial and dynamic boundary conditions are from a global simulation with $4° \times 5°$ resolution. GEOS-Chem uses operator splitting to separately solve the equations for transport and chemistry. The chemical time-step is 10 minutes at $0.25° \times 0.3125°$ resolution, 30 minutes at $2° \times 2.5°$, and 60 minutes at $4° \times 5°$. The transport time-step is half the chemical time-step. Boundary layer mixing depths is scaled down from the GEOS values by 40% to match observations from aircraft lidar during SEAC[4]RS (Zhu et al., 2015).

The SEAC[4]RS simulation is based on GEOS-Chem version 9-02 (http://wiki.seas.harvard.edu/geos-chem/index.php/GEOS-Chem_v9-02) including a detailed mechanism for $HO_x$-$NO_x$-VOC-$O_3$-bromine-aerosol tropospheric chemistry with 196 chemical species (Parrella et al., 2012; Mao et al., 2013). A number of updates to isoprene chemistry were implemented as described by Fisher et al. (2015), Marais et al. (2015), and Travis et al. (2015). US anthropogenic emissions are from the 2011 National





Emissions Inventory (NEI), with national scaling for 2013 and 60% downward correction for $NO_x$ as described in Travis et al. (2015). Biogenic emissions are from the Model of Emissions of Gases and Aerosols from Nature (MEGANv2.1) (Guenther et al., 2012) with 15% downward correction for isoprene (Zhu et al., 2015). Lightning $NO_x$ emissions are constrained by satellite observations as described in Murray et al. (2012). Soil $NO_x$ emissions, including fertilizer, are from Hudman et al. (2012)

with Jacob and Wofsy (1990) canopy reduction factors.

Figure 1 shows the emissions of $NO_x$ and isoprene over the Southeast US domain at $0.25° \times 0.3125°$ resolution. In this domain, isoprene emission is mainly from forests and $NO_x$ emission largely follows population and power plants. The resulting spatial segregation between isoprene and $NO_x$ emissions has important implications for whether isoprene oxidation takes place by high-$NO_x$ or low-$NO_x$ pathways. We ensure that emission totals are the same at all model resolutions so that they are not a

factor of differences in results. This required minor scaling of natural VOC emissions that depend on environmental variables.

Comparisons between model and aircraft observations use model output sampled along the DC-8 flight tracks (Figure 1), and observations from a 60-s merged data set. For computing aggregate statistics, such as probability distributions, we use observations directly from the 60-s data set. For computing correlations between model and observations (Figure 5), we average data over model grid cells along individual flight tracks. We focus on daytime continental data over the Southeast US domain

($94.5° - 75°W$, $29.5° - 40°N$, box in Figure 1) and within the mixed layer defined along the flight track from lidar measurements (Hair et al., 2008). Typical mixed layer heights ($10^{th}$ and $90^{th}$ percentiles) during the campaign ranged from 600 m to 2200 m, with a mean value of 1500 m. The aircraft occasionally targeted fire plumes and we remove those as diagnosed by measured acetonitrile concentrations exceeding 225 ppt. Our analysis domain also excludes the Houston plume, targeted on two SEAC[4]RS flights (Sep 16 and 18) and for which our model would not be expected to give a representative simulation due

to potentially large highly reactive petrochemical emissions. Unlike Travis et al. (2015), we do not remove urban plumes as diagnosed by $NO_2$ concentrations exceeding 8 ppb.

## 3   Prevalence of high-$NO_x$ and low-$NO_x$ isoprene oxidation pathways

Oxidant chemistry over the Southeast US in summer is largely determined by the interactions between isoprene and $NO_x$. Isoprene oxidation by the high-$NO_x$ pathway produces ozone and also organic nitrates that act as either sinks or reservoirs for

$NO_x$, while oxidation by the $HO_2$ pathway tends to scavenge $HO_x$ radicals. The different pathways also lead to formation of different types of secondary organic aerosol, with higher yields by the $HO_2$ pathway (Marais et al., 2015).

Travis et al. (2015) diagnosed the prevalence of different isoprene oxidation pathways in the GEOS-Chem ($0.25° \times 0.3125°$) simulation of the Southeast US during SEAC[4]RS on the basis of the fraction of the first-generation $ISOPO_2$ radicals that react with NO. They found on average that 56% of $ISOPO_2$ radicals reacted by the high-$NO_x$ pathway. The low-$NO_x$ pathways

mostly involved reaction of $ISOPO_2$ with $HO_2$ (25%), producing isoprene hydroperoxides (ISOPOOH), and minor channels from $ISOPO_2$ isomerization (15%) and reactions with other organic peroxy radicals (4%). The transition from high- to low-$NO_x$ pathways occurred at a NO concentration of about 60 ppt, corresponding to a $NO_x$ concentration of about 300 ppt.





The effect of the geographical segregation between $NO_x$ and isoprene emissions is illustrated in Figure 2, which shows the relationship between $NO_x$ and isoprene concentrations along the flight tracks. $NO_x$ was measured by chemiluminescence (Ryerson et al., 2000) and isoprene by proton-transfer-reaction mass spectrometry (PTR-MS) (Hansel et al., 1999). The observations show a negative correlation that is captured in the simulation at $0.25° \times 0.3125°$ resolution but not at coarser resolutions. The $4° \times 5°$ simulation actually shows positive correlations, likely reflecting a common sensitivity to regional temperature and stagnation.

For the SEAC[4]RS data set sampled along the flight tracks, the $0.25° \times 0.3125°$ simulation finds that 42% of the $ISOPO_2$ radicals react with NO. That fraction increases to 44% for the $2° \times 2.5°$ simulation and 49% for the $4° \times 5°$ simulation. Over the whole Southeast US domain (box in Figure 1), the fraction of $ISOPO_2$ radicals reacting by the high-$NO_x$ pathway is 54% at $0.25° \times 0.3125°$ resolution and 59% at the $2° \times 2.5°$ and $4° \times 5°$ resolutions. Our values differ slightly from Travis et al. (2015) because we did not exclude ocean grid cells in the domain in order to keep comparisons between different resolutions consistent. There is more variability at the $0.25° \times 0.3125°$ resolution, with the high-$NO_x$ pathway dominating in 55% of grid cells averaged over August and September 2013 vs. 74% at both $2° \times 2.5°$ and $4° \times 5°$. Still, we find that the segregation between the different pathways is dampened relative to the segregation between $NO_x$ and isoprene emissions. This is due to OH depletion under low-$NO_x$ conditions prolonging the lifetime of isoprene and its oxidation products, allowing them to travel to higher-$NO_x$ environments before being oxidized.

The SEAC[4]RS aircraft payload included measurements of first-generation isoprene nitrates (ISOPN), a product of the high-$NO_x$ pathway, and ISOPOOH, the principal product of the $HO_2$ pathway (Crounse et al., 2006). Figure 3 shows a cumulative probability distribution function (CDF) plot of the [ISOPN]/([ISOPOOH] + [ISOPN]) ratio in the Southeast US mixed layer, providing a diagnostic of whether isoprene oxidation proceeds by the high-$NO_x$ pathway (high ratio) or the low-$NO_x$ pathways (low ratio). We see that the $0.25° \times 0.3125°$ simulation is much more consistent with observations than the $2° \times 2.5°$ and $4° \times 5°$ simulations. Although the ratios are sensitive to ISOPN and ISOPOOH lifetimes, and model error in those lifetimes would affect the ratio, the increased variance in the $0.25° \times 0.3125°$ model clearly demonstrates better partitioning between the different pathways.

## 4   Chemical variability and bias

Figure 4 shows the simulated and observed CDFs of $NO_x$, isoprene, formaldehyde, and ozone concentrations sampled along the flight tracks in the Southeast US mixed layer. Formaldehyde was measured by Compact Atmospheric Multispecies Spectrometer (CAMS) (Richter et al., 2015) and ozone by chemiluminescence (Ryerson et al., 2000). $NO_x$ and isoprene are primary (directly emitted) and have mean lifetimes against chemical loss of a few hours and less than an hour, respectively. The $NO_x$ distribution is approximately lognormal while isoprene is better described by a Weibull distribution. Formaldehyde and ozone are secondary (chemically produced) and their distributions are more normal. Most of the formaldehyde in SEAC[4]RS originated from isoprene oxidation (Zhu et al., 2015).





We find that, with the exception of formaldehyde, differences between the different model resolutions are mainly at the high tails of the distributions. In the case of $NO_x$, the highest resolution model better captures the high tail in the observations due to urban plumes. In the case of isoprene the highest resolution model over-predicts the high tail in the observations, which could reflect errors in the fine structure of MEGAN emissions or excessive local depletion of OH (the main isoprene sink) as a result of high isoprene. In the case of ozone, the highest resolution model only marginally improves the simulation of the high

tail in the observations (industrial plumes downwind of Port Arthur, Texas), for which even $0.25° \times 0.3125°$ may not provide adequate resolution. In the case of formaldehyde, the highest resolution model actually produces weaker maxima that are more consistent with observations. Production of formaldehyde from isoprene oxidation has higher yield in the high-$NO_x$ pathway than the low-$NO_x$ pathways (Marais et al., 2012). High isoprene is associated with low $NO_x$ in the observations (Figure 2) and therefore with a low formaldehyde yield but this is captured only at the highest model resolution. This result would have

implications for the generally assumed linear relationship used to infer isoprene emissions from satellite measurements of formaldehyde columns (Palmer et al., 2003).

These high tails aside, the bulk of the distributions shows very little difference between model results at different resolutions, despite $NO_x$ levels spanning 4 orders of magnitude. This suggests that the coarser resolutions are adequate for simulating regional averages across the domain studied. An exception is formaldehyde, where the higher resolution better reproduces the

observations above the median. As pointed out above, formaldehyde is particularly sensitive to the separation between the high-$NO_x$ pathway and low-$NO_x$ pathways.

One important caveat is that although the Southeast US represents a variety of different chemical environments, we only explored non-linear effects down to the scale of $0.25° \times 0.3125°$ ($\approx 28$ km). Features below that scale have been shown to have strong gradients that can result in biases. Investigation of non-linear effects on the scale of a few kilometers has demonstrated

strong resolution effects for highly urbanized regions (Shrestha et al., 2009; Tie et al., 2010), power plant or industrial plumes (Valin et al., 2011; Henderson et al., 2010), and ship plumes (Davis et al., 2001; Charlton-Perez et al., 2009; Vinken et al., 2011). Nevertheless, the general ability of the GEOS-Chem simulation to capture the variance in the SEAC[4]RS observations implies that these small-scale effects are not dominant at least for the Southeast US.

Figure 5 presents a Taylor Diagram of modeled $NO_x$, isoprene, formaldehyde, and ozone concentration statistics compared

to observations. The Taylor Diagram is a concise graphical summary of how well two patterns match each other in terms of their correlation, root-mean-squared difference, and variances (Taylor, 2001). Comparison of variances for different resolutions is consistent with the previously discussed information from Figure 4. Correlations with observations improve when resolution is increased from $4° \times 5°$ to $2° \times 2.5°$, but then generally degrade at $0.25° \times 0.3125°$ (except for $NO_x$). This is because finer-scale features are more difficult to model (Kiley et al., 2003; Mathur et al., 2005; Arunachalam et al., 2006; Valari and Menut, 2008).

A higher resolution model will be penalized for placing these features in the wrong place or time, while a coarse resolution model does not attempt to resolve them.





## 5 Implications for global influence

From a global modeling perspective, the ability to simulate local extrema in a chemical source region is generally not critical and the focus instead is on simulation of regional means and export to the global atmosphere. From that standpoint, the general insensitivity of concentrations to model resolution over the bulk of the distributions suggests that coarse resolution is adequate for global modeling purposes. We confirm this result by examination of the regional budgets of total reactive nitrogen oxides

($NO_y \equiv NO_x$ + oxidation products) and ozone over the Southeast US at different model resolutions.

Figure 6 shows mean simulated and observed daytime concentrations of $NO_y$ species in the Southeast US mixed layer during SEAC[4]RS. Total $NO_y$ in the model is within 10% of observations, at all resolutions, but there is somewhat more difference in the speciation of $NO_x$ oxidation products. The $0.25° \times 0.3125°$ simulation has lower peroxyacetylnitrate (PAN) than the coarser resolutions, in better agreement with observations, due to less artificial mixing of $NO_x$ and isoprene emissions

as previously pointed out by Ito et al. (2009). All model resolutions show similar low bias relative to observed total alkyl nitrates ($\Sigma$ANs). This low bias is discussed in Fisher et al. (2015). The model has larger nitric acid concentrations at higher resolution because it resolves better the positive correlation between $NO_x$ and OH concentrations. Artificial mixing of $NO_x$ with isoprene in the coarser-resolution models slows down oxidation by OH to nitric acid (since isoprene provides a sink for OH) and increases the production of organic nitrates. The $0.25° \times 0.3125°$ model has a 17% high bias relative to observed nitric

acid from the SAGA instrument shown in Figure 6, and the bias would be larger relative to nitric acid measurements from the Caltech Chemical Ionization Mass Spectrometer also flown on the aircraft. We find that the regional mean $NO_x$ lifetime with respect to conversion to nitric acid is 0.66 days for the $4° \times 5°$ model, 0.62 days for the $2° \times 2.5°$ model, and 0.51 days for the $0.25° \times 0.3125°$ model. Although there is no significant change in the simulated regional mean OH concentrations across different resolutions, the higher resolution model captures better the association of elevated OH with $NO_x$.

From a global model perspective, chemical venting from continental source regions such as the Southeast US is of paramount importance. Figure 7 shows CDF plots for ozone and $NO_x$ in the lower free troposphere (from mixed layer top to 4 km altitude). Model results are insensitive to resolution, with differences even smaller than in the mixed layer. As in the mixed layer, all model resolutions simulate the median of the distributions well, and differences across model resolution occur mainly at the tails of the distributions. Overall, we find that grid resolution has little effect on the export of ozone and its precursors out of

the mixed layer.

## 6 Conclusions

Production of ozone and organic aerosol in continental atmospheres is highly sensitive to whether isoprene emitted by vegetation is oxidized by the high-$NO_x$ pathway (where peroxy radicals react with NO) or by the low-$NO_x$ pathways (where peroxy radicals react mostly with $HO_2$). This distinction between pathways is becoming increasingly relevant in the US as anthro-

pogenic $NO_x$ emissions decrease. In this work, we used SEAC[4]RS aircraft observations in the mixed layer over the Southeast US to test the ability of the GEOS-Chem chemical transport model (CTM) at different horizontal resolutions ($0.25° \times 0.3125°$, $2° \times 2.5°$, $4° \times 5°$ ) to simulate the different pathways and the resulting variability in concentrations. $2° \times 2.5°$ and $4° \times 5°$





are the standard resolutions used in global GEOS-Chem simulations of tropospheric chemistry while the $0.25° \times 0.3125°$ continental-scale resolution is a new GEOS-Chem capability developed for interpretation of the SEAC[4]RS data.

Emissions of $NO_x$ and isoprene tend to be spatially segregated. SEAC[4]RS observations in the mixed layer show a negative correlation between isoprene and $NO_x$ concentrations that is captured in the model at $0.25° \times 0.3125°$ resolution but not at coarser resolution. In the $0.25° \times 0.3125°$ resolution model, 54% of isoprene oxidation in the Southeast US takes place by the high-$NO_x$ pathway compared to 59% at the coarser resolutions. Observed ratios of isoprene nitrates (ISOPN) to isoprene hydroperoxides (ISOPOOH) show segregation between high- and low-$NO_x$ pathways that is much better captured at $0.25° \times 0.3125°$ than at coarser resolutions. The segregation between high- and low-$NO_x$ pathways is less than would be expected from the segregation of $NO_x$ and isoprene emissions because OH depletion in low-$NO_x$ environments allows isoprene to travel to higher-$NO_x$ environments to become oxidized.

We examined the ability of the model at different resolutions to simulate the observed probability distributions of $NO_x$, isoprene, formaldehyde, and ozone concentrations across the Southeast US. Differences between model resolutions are mainly at the high tails of the distributions. There is remarkably little difference for the bulk of the distributions. An exception is formaldehyde, which is overestimated at coarser model resolution as more isoprene is oxidized by the high-$NO_x$ pathway, producing more formaldehyde. Isoprene oxidation is more likely to proceed by the low-$NO_x$ pathways where isoprene emissions are high, and this has important implications for inferring isoprene emissions from satellite observations of formaldehyde columns.

Spatial correlations between model and observations improve as resolution increases from $4° \times 5°$ to $2° \times 2.5°$ but then decrease as resolution increases further to $0.25° \times 0.3125°$. This shows that finer modes of variability are more difficult to capture in models of commensurate resolution.

Increasing model resolution leads to faster conversion of $NO_x$ to nitric acid, because OH correlates positively with $NO_x$, while slowing down conversion to organic nitrates. However, the effect on the mean budget of reactive nitrogen oxides ($NO_y$) in the mixed layer is small. Furthermore, comparisons of $NO_x$ and ozone concentrations in the lower free troposphere indicate no significant sensitivity to model resolution. Although higher resolution models are necessary for certain purposes, such as examining the tails of distributions and for studying the relative importance of different chemical pathways, the overall relative insensitivity of oxidant chemistry to model resolution in the challenging environment of the Southeast US suggests that mean regional boundary layer chemistry can be simulated adequately at coarse resolution (such as $2° \times 2.5°$) for global modeling purposes.

*Acknowledgements.* We are grateful to the entire NASA SEAC[4]RS team for their help in the field. This work was funded by the NASA Atmospheric Composition Modeling and Analysis Program and by the NASA Tropospheric Chemistry Program. JAF acknowledges financial support from a University of Wollongong Vice Chancellor's Postdoctoral Fellowship. Isoprene measurements during SEAC[4]RS were supported by the Austrian Federal Ministry for Transport, Innovation and Technology (bmvit) through the Austrian Space Applications Programme (ASAP) of the Austrian Research Promotion Agency (FFG). AW and TM received support from the Visiting Scientist Program at the National Institute of Aerospace (NIA).





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



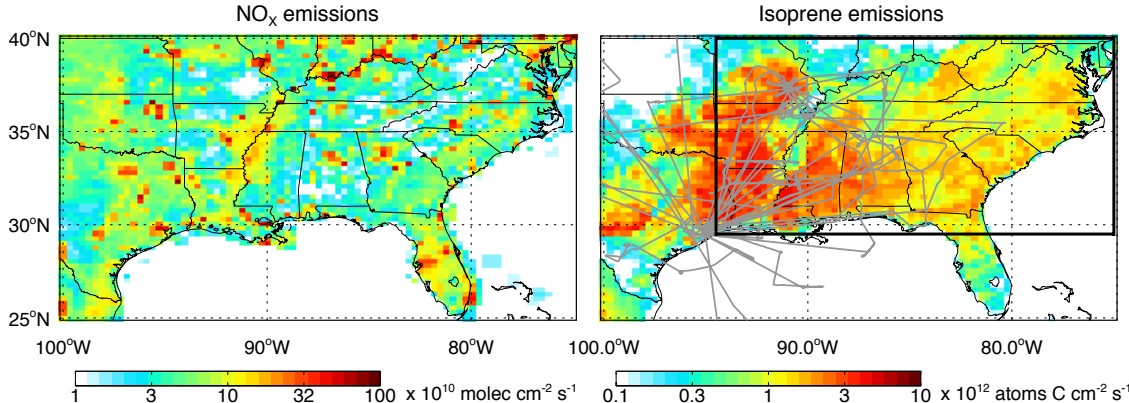

**Figure 1.** GEOS-Chem surface emissions of $NO_x$ and isoprene in August 2013 at $0.25° \times 0.3125°$ resolution. See text for emission inventory references. $NO_x$ emissions shown here include contributions from combustion and soils but not lightning. Grey lines on isoprene panel indicate flight tracks of the DC-8 during the SEAC$^4$RS campaign. The black line delineates the Southeast US as used for regional budget analyses in the text.

Wild, O. and Prather, M. J.: Global tropospheric ozone modeling: Quantifying errors due to grid resolution, J. Geophys. Res., 111, doi:10.1029/2005jd006605, 2006.

Yamaji, K., Ikeda, K., Irie, H., Kurokawa, J., and Ohara, T.: Influence of model grid resolution on $NO_2$ vertical column densities over East Asia, J. Air Waste Manage., 64, 436–444, doi:10.1080/10962247.2013.827603, 2014.

Zhang, L., Jacob, D. J., Downey, N. V., Wood, D. A., Blewitt, D., Carouge, C. C., van Donkelaar, A., Jones, D. B. A., Murray, L. T., and
5 Wang, Y.: Improved estimate of the policy-relevant background ozone in the United States using the GEOS-Chem global model with $1/2° \times 2/3°$ horizontal resolution over North America, Atmos. Environ., 45, 6769–6776, doi:10.1016/j.atmosenv.2011.07.054, 2011.

Zhu, L., Jacob, D. J., Mickley, L. J., Kim, P. S., Fisher, J. A., Travis, K., Yu, K., Yantosca, R. M., Sulprizio, M. P., Fried, A., Hanisco, T., Wolfe, G., Abad, G. G., Chance, K., De Smedt, I., and Yang, K.: Observing atmospheric formaldehyde (HCHO) from space: validation of six operational products with extensive aircraft observations, in preparation, 2015.





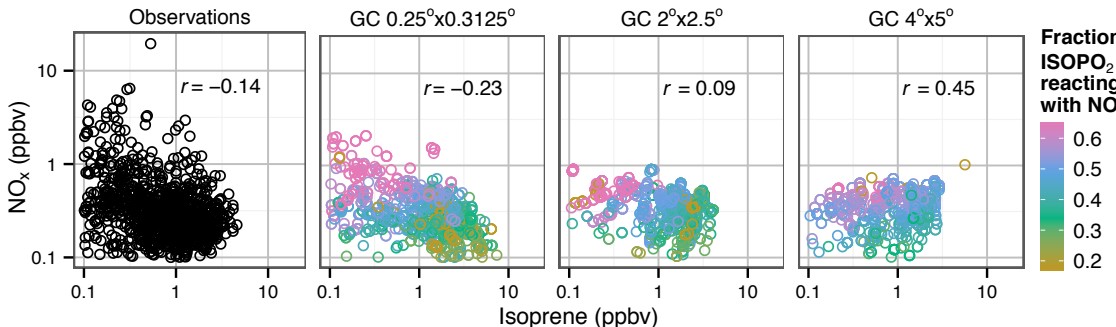

**Figure 2.** Relationship between $NO_x$ and isoprene concentrations in the mixed layer in $SEAC^4RS$. Observations (60-s average) are compared to the GEOS-Chem (GC) simulations at different resolutions sampled along the flight tracks over the Southeast US domain of Figure 1. Pearson's correlation coefficients ($r$) are inset. The model panels are colored by the fraction of the first-generation isoprene peroxy radical ($ISOPO_2$) reacting with NO. A fraction less than 0.5 indicates that low-$NO_x$ pathways dominate for isoprene oxidation.

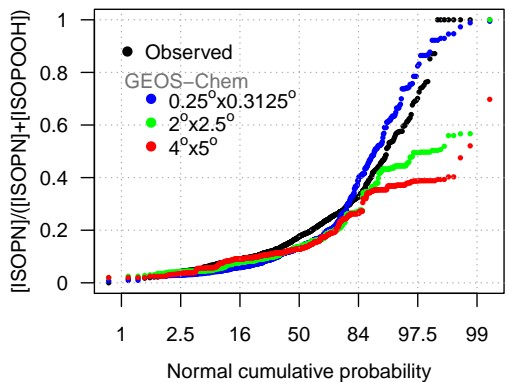

**Figure 3.** Cumulative probability distribution functions (CDFs) of the $\dfrac{[ISOPN]}{[ISOPN] + [ISOPOOH]}$ ratio measuring the relative importance of the high-$NO_x$ pathway vs. low-$NO_x$ pathways for isoprene oxidation. High values of the ratio indicate a dominant high-$NO_x$ pathway. Mixed layer observations from the $SEAC^4RS$ aircraft over the Southeast US are compared to GEOS-Chem output along the flight tracks at different resolutions. The $x$-axis is a normal probability scale such that a normal distribution would plot on a straight line.





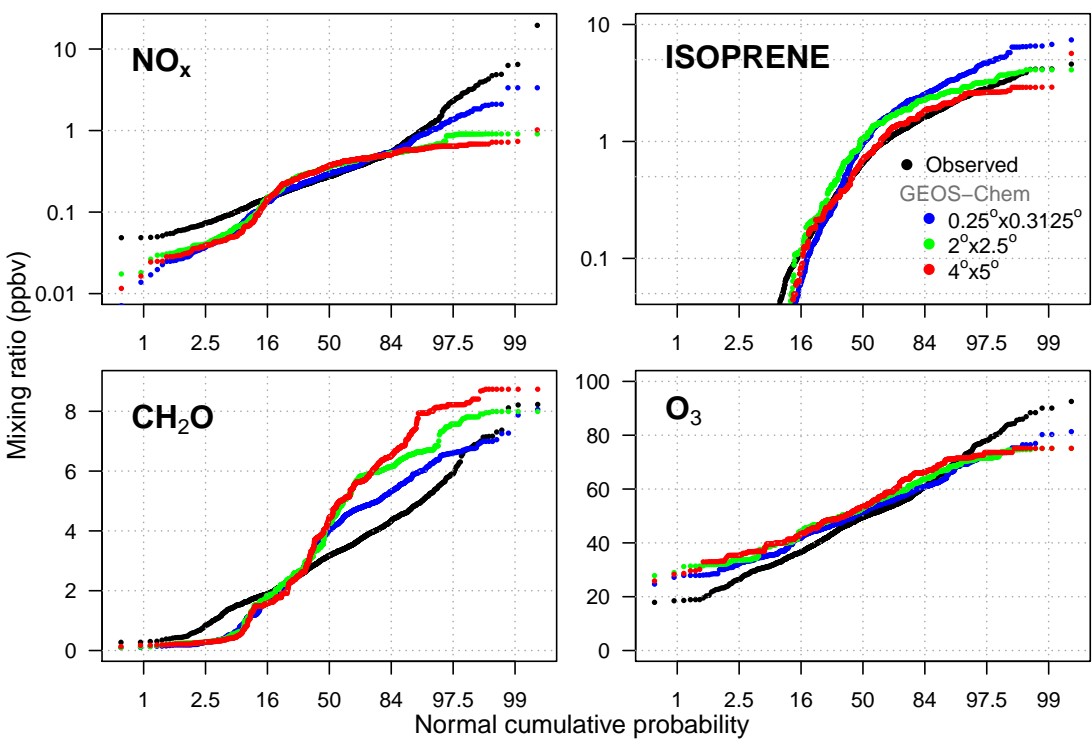

**Figure 4.** Cumulative probability distribution functions (CDFs) of $NO_x$, isoprene, formaldehyde ($CH_2O$), and ozone ($O_3$) concentrations in the Southeast US mixed layer during the SEAC[4]RS aircraft campaign. Observations are compared to GEOS-Chem model values at different resolutions ($0.25° \times 0.3125°$, $2° \times 2.5°$, $4° \times 5°$) sampled along the flight tracks over the Southeast US domain of Figure 1. The $x$-axis is a normal probability scale such that a normal distribution (formaldehyde, ozone) or lognormal distribution ($NO_x$, isoprene) would plot on a straight line.



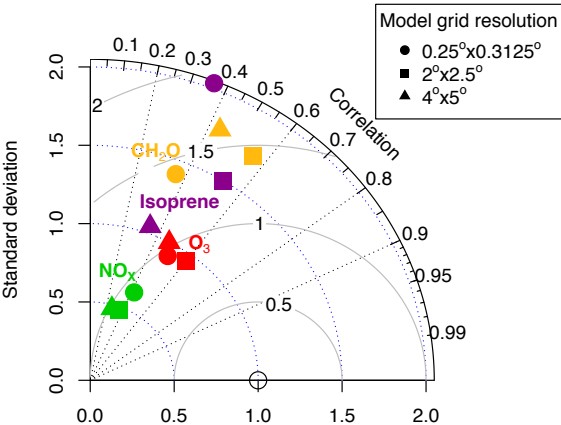

**Figure 5.** Taylor Diagram for $NO_x$, isoprene, formaldehyde, and ozone concentration statistics sampled in the Southeast US mixed layer along SEAC[4]RS flight tracks (Figures 1 and 4). GEOS-Chem model results at different grid resolutions are compared to observations at the corresponding resolutions. Standard deviation is plotted along the radial coordinate, while the angular coordinate denotes the Pearson's correlation coefficient between model and observations. Model standard deviation is normalized to the observations, so that a value above 1 indicates greater variance than observed. The open circle located at (1, 1) represents the observations. The grey arcs represent the root-mean-squared error between model and observations after the mean bias has been removed.

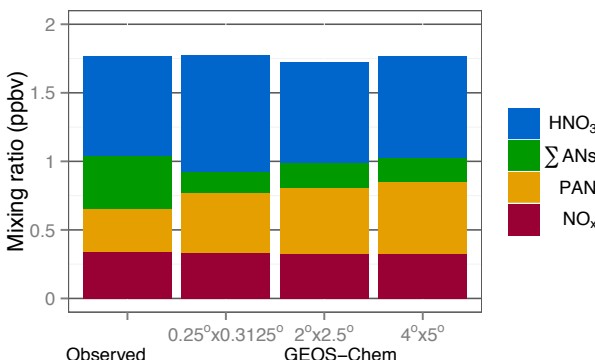

**Figure 6.** Mean concentrations of reactive nitrogen oxides ($NO_y$) species over the Southeast US during SEAC[4]RS. Aircraft observations in the mixed layer over the Southeast US domain of Figure 1 are compared to GEOS-Chem model values at different resolutions. ΣANs refers to the sum of alkyl and multifunctional nitrates and PAN to peroxyacetylnitrate. Observations are from T. Ryerson for $NO_x$, R. Cohen for ΣANs, G. Huey for PAN, and J. Dibb for nitric acid ($HNO_3$) (Toon et al., 2015).





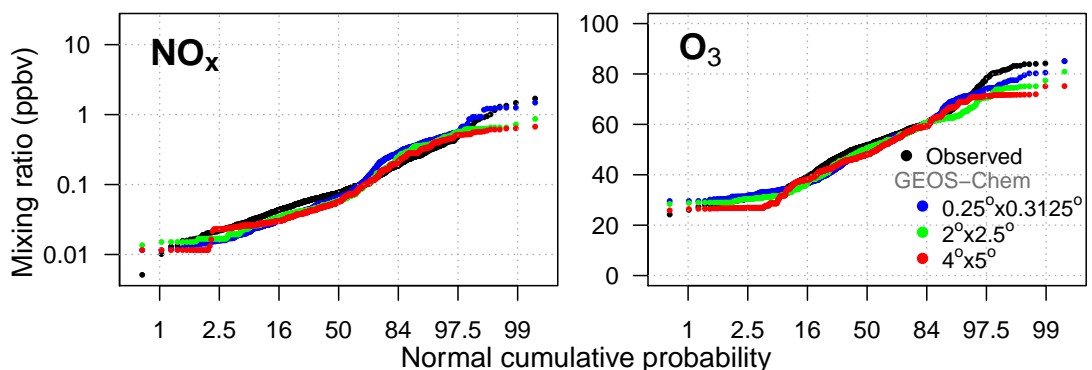

**Figure 7.** Cumulative probability distribution functions (CDFs) of $NO_x$ and ozone ($O_3$) concentrations in the lower free troposphere over the Southeast US (from mixed layer top to 4 km altitude) during the SEAC[4]RS aircraft campaign. Observations are compared to GEOS-Chem model values at different resolutions sampled along the flight tracks over the Southeast US domain of Figure 1. The $x$-axis is a normal probability scale such that a normal distribution (ozone) or lognormal distribution ($NO_x$) would plot as a straight line.