# Peer review of "Sensitivity to grid resolution in the ability of a chemical transport model to simulate observed oxidant chemistry under high-isoprene conditions"

_Atmospheric Chemistry and Physics, 2015_

## Referee Comment (RC1) · Anonymous Referee #1 · 12 Feb 2016

The manuscript by Yu et al. is a valuable contribution to a long-term discussion on the role of spatial resolution in determining the total error of an atmospheric chemistry-transport model. In brief, the authors focus on the analysis of an aircraft campaign in the South-Eastern US, which is a region characterized by large biogenic isoprene emissions, and compare the observations with simulations carried out with the same model (GEOS-Chem) at three spatial resolution (increasing from c.a. 400 to c.a. 28 km). They found a little impact by increased resolution on simulated values of ozone and NOx, especially in the free troposphere. They thus conclude that a coarse resolution model is adequate to resolve regional contributions to ozone export on a global perspective.
[Figure]

I found the manuscript concise, clear and well written, and I tend to agree with the main conclusions of the authors, based on the material showed here. I only have a few comments, which are more requests of clarification from an interested reader, that may further improve the interpretation of some of the results:

- page 2, lines 19-29: in this paragraph the authors review previous similar studies on the horizontal resulution issue, but at the end of the manuscript it is not clear to the reader what is the advancement/difference/similarity (if any) with these studies. I recommend inclusing a short paragraph in the conclusions on that.

- page 5, lines 4-5: the varying ISOP-NOx relationship in the model is one of the most intriguing results presented in the manuscript. However, its interpretation is left too much to speculations by the reader. Can this analysis be improved? For example, can the referred statement on "temperature and stagnation" be proved calculating correlatons with temperature and a stagnation index?

- page 5, lines 13-15: The role of OH is recurring here and throughout the manuscript. Please consider to include some visualization of the changing OH or related species fields, this may help the reader in the interpretation of results.

- page 6, lines 26-27: I do not completely agree with this statement, in particular for isoprene. Looking at Figure 4, all the model realizations look pretty similar to observations, however Taylor diagram in Figure 5 display a dramatic decrease of model skills with increasing resolution. It is not clear to the reader why it happens at this point, it comes as some sort of surprise, so it needs further analysis and discussion. Perhaps, here and for other purposes it would be beneficial to include some sort of more direct visualization of the model-to-obs comparisons (e.g. simple timeseries of data, or scatter plots), maybe in the supplement, in order to keep the main text concise.
* * *

---

## Referee Comment (RC2) · Anonymous Referee #2 · 14 Mar 2016

The authors explain and show concisely the main thesis of the manuscript which falls within the scope of the journal. References and discussion of previous work is sufficient. I recommend publication with minor revisions.

Main comments

Many aspects (mostly chemical) are left to other papers which are not all available in ACPD yet (e.g. Travis et al.). Therefore, I would like to see in the manuscript the relative differences for the simulated daytime ozone levels going from $2°x2.5°$ to $0.25°x0.3125°$ resolution. Such a figure for August at the surface and at 4 km would be sufficient and help make the point of the authors clearer.

[Figure]

The authors discuss the segregation of isoprene and NOx emissions and of isoprene oxidation pathways. These statements need some abstraction by the reader and are obvious. They should be substantiated by calculations. The literature I am aware of discusses the intensity of chemical segregation between two species like isoprene and OH and not between two pathways or emissions. Please add mathematical definitions, calculations and a figure for different model resolutions.

---

## Author Comment (AC1) · 26 Mar 2016

Reviewer comments are in red. Responses are in black, with changes to the text in italics.

**1   Responses to Reviewer 1**

The manuscript by Yu et al. is a valuable contribution to a long-term discussion on the role of spatial resolution in determining the total error of an atmospheric chemistry transport model. In brief, the authors focus on the analysis of an aircraft campaign

in the South-Eastern US, which is a region characterized by large biogenic isoprene emissions, and compare the observations with simulations carried out with the same model (GEOS-Chem) at three spatial resolution (increasing from c.a. 400 to c.a. 28 km). They found a little impact by increased resolution on simulated values of ozone and $NO_x$, especially in the free troposphere. They thus conclude that a coarse resolution model is adequate to resolve regional contributions to ozone export on a global perspective.

I found the manuscript concise, clear and well written, and I tend to agree with the main conclusions of the authors, based on the material showed here. I only have a few comments, which are more requests of clarification from an interested reader, that may further improve the interpretation of some of the results:

**1.1 Specific comments**

page 2, lines 19-29: in this paragraph the authors review previous similar studies on the horizontal resulution issue, but at the end of the manuscript it is not clear to the reader what is the advancement/difference/similarity (if any) with these studies. I recommend inclusing a short paragraph in the conclusions on that.

We have updated the abstract, body of text, and conclusions to reflect these concerns.

In the abstract, the following text has been added:
*The good agreement of simulated and observed concentration variances implies that smaller-scale non-linearities (urban and power plant plumes) are not important on the regional scale.*

In the body of the text, on line 14 of page 6, we add the following:
*Non-linear effects have been reported in previous model studies at the km-scale of urban areas (Kumar and Russell, 1996; Chock et al., 2002; Shrestha et al., 2009; Tie et al., 2010) or pollution plumes (Henderson et al., 2010; Valin et al., 2011), and these*

*small-scale effects are not resolved here even at* $0.25^o \times 0.3125^o$ *resolution.*

In the conclusions, we amend the following paragraph (addition in italics):
Spatial correlations between model and observations improve as resolution increases from $4^o \times 5^o$ to $2^o \times 2.5^o$ but then decrease as resolution increases further to $0.25^o \times 0.3125^o$. This shows that finer modes of variability are more difficult to capture in models of commensurate resolution, *consistent with the results of previous studies such as Fiore et al. (2003) and Valari and Menut (2008).*

And add the following sentence:
*Previous model studies have pointed out significant chemical non-linearities at the scale of urban and industrial plumes, smaller than the $0.25^o \times 0.3125^o$ resolution used here. However, the ability of GEOS-Chem to simulate the variability of concentrations observed in SEAC[4]RS implies that such small-scale effects are not important on the regional scale.*

page 5, lines 4-5: the varying ISOP-NO$_x$ relationship in the model is one of the most intriguing results presented in the manuscript. However, its interpretation is left too much to speculations by the reader. Can this analysis be improved? For example, can the referred statement on "temperature and stagnation" be proved calculating correlatons with temperature and a stagnation index?

We now compute correlations with temperature and update line 5 on page 5 to read as follows:
The $4^o \times 5^o$ simulation actually shows positive correlations, likely reflecting a common sensitivity to *regional stagnation and resulting high temperatures. We find in the model that NO$_x$ concentrations are strongly correlated with surface air temperature on the $4^o \times 5^o$ scale ($r = 0.66$) but not on the $0.25^o \times 0.3125^o$ scale ($r = 0.17$).*

page 5, lines 13-15: The role of OH is recurring here and throughout the manuscript. Please consider to include some visualization of the changing OH or related species fields, this may help the reader in the interpretation of results.

We have updated Figure 2 to include panels colored by OH to show more clearly OH depletion under low-NO$_x$ conditions.

page 6, lines 26-27: I do not completely agree with this statement, in particular for isoprene. Looking at Figure 4, all the model realizations look pretty similar to observations, however Taylor diagram in Figure 5 display a dramatic decrease of model skills with increasing resolution. It is not clear to the reader why it happens at this point, it comes as some sort of surprise, so it needs further analysis and discussion. Perhaps, here and for other purposes it would be beneficial to include some sort of more direct visualization of the model-to-obs comparisons (e.g. simple timeseries of data, or scatter plots), maybe in the supplement, in order to keep the main text concise

We amended the text to clarify that correlations are computed using observations averaged to model grid and timestep. We average observations to model grid and timestep as this is a fairer way to compare model output to observations when computing correlations, but it also means correlations will worsen at higher resolution because higher resolution features are inherently more difficult to capture.

**2 Responses to Reviewer 2**

The authors explain and show concisely the main thesis of the manuscript which falls within the scope of the journal. References and discussion of previous work is sufficient. I recommend publication with minor revisions.

**2.1 Main comments**

[Figure]

Many aspects (mostly chemical) are left to other papers which are not all available in ACPD yet (e.g. Travis et al.). Therefore, I would like to see in the manuscript the relative differences for the simulated daytime ozone levels going from $2^o \times 2.5^o$ to $0.25^o \times 0.3125^o$

resolution. Such a figure for August at the surface and at 4 km would be sufficient and help make the point of the authors clearer.

Travis et al., (2016) and the other GEOS-Chem papers referenced are now either published or in ACPD. References have been updated in our manuscript.

The authors discuss the segregation of isoprene and $NO_x$ emissions and of isoprene oxidation pathways. These statements need some abstraction by the reader and are obvious. They should be substantiated by calculations. The literature I am aware of discusses the intensity of chemical segregation between two species like isoprene and OH and not between two pathways or emissions. Please add mathematical definitions, calculations and a figure for different model resolutions.

Additional discussion of isoprene oxidation pathways can be found in Travis et al., (2016).

We have updated Figure 2 to include panels colored by OH to show more clearly OH depletion under low-$NO_x$ conditions (see attached figure).

**3   References**

Chock, D. P., Winkler, S. L., and Sun, P.: Effect of grid resolution and subgrid assumptions on the model prediction of a reactive bouyant plume under convective conditions, Atmos. Environ., 36, 4649–4662, doi:10.1016/S1352-2310(02)00422-3, 2002.

Fiore, A. M., Jacob, D. J., Mathur, R., and Martin, R. V.: Application of empirical orthogonal functions to evaluate ozone simulations with regional and global models, J. Geophys. Res., 108, doi:10.1029/2002jd003151, 4431, 2003.

Henderson, B. H., Jeffries, H. E., Kim, B.-U., and Vizuete, W. G.: The influence of model resolution on ozone in industrial volatile organic compound plumes, J. Air Waste

Manage., 60, 1105–1117, doi:10.3155/1047-3289.60.9.1105, 2010.

Kumar, N. and Russell, A. G.: Multiscale air quality modeling of the northeastern United States, Atmos. Environ., 30, 1099–1116, doi:10.1016/1352-2310(95)00317-7, 1996.

Shrestha, K. L., Kondo, A., Kaga, A., and Inoue, Y.: High-resolution modeling and evaluation of ozone air quality of Osaka using MM5-CMAQ system, J. Environ. Sci., 21, 782–789, doi:10.1016/s1001-0742(08)62341-4, 2009.

Tie, X., Brasseur, G., and Ying, Z.: Impact of model resolution on chemical ozone formation in Mexico City: application of the WRF-Chem model, Atmos. Chem. Phys., 10, 8983–8995, doi:10.5194/acp-10-8983-2010, 2010.

Travis, K. R., Jacob, D. J., Fisher, J. A., Kim, P. S., Marais, E. A., Zhu, L., Yu, K., Miller, C. C., Yantosca, R. M., Sulprizio, M. P., Thompson, A. M., Wennberg, P. O., Crounse, J. D., St. Clair, J. M., Cohen, R. C., Laugher, J. L., Dibb, J. E., Hall, S. R., Ullmann, K., Wolfe, G. M., Pollack, I. B., Peischl, J., Neuman, J. A., and Zhou, X.: $NO_x$ emissions, isoprene oxidation pathways, vertical mixing, and implications for surface ozone in the Southeast United States, Atmos. Chem. Phys. Discussions, doi:10.5194/acp-2016-110, 2016.

Valari, M. and Menut, L.: Does an increase in air quality models' resolution bring surface ozone concentrations closer to reality?, J. Atmos. Ocean Tech., 25, 1955–1968, doi:10.1175/2008jtecha1123.1, 2008.

Valin, L. C., Russell, A. R., Hudman, R. C., and Cohen, R. C.: Effects of model resolution on the interpretation of satellite $NO_2$ observations, Atmos. Chem. Phys., 11, 11 647–11 655, doi:10.5194/acp-11-11647-2011, 2011.
* * *
[Figure]

[Figure]

**Fig. 1.** Updated Figure 2 now includes three model panels colored by OH with regional mean OH (computed along flight tracks) inset.

---

## Author Response (AR1)

**RESPONSE TO REVIEWERS**

Ms. Ref. No.: acp-2015-980

Title: Sensitivity to grid resolution in the ability of a chemical transport model to simulate observed oxidant chemistry under high-isoprene conditions

Journal: Atmos. Chem. Phys. Discuss.

Reviewer comments are in red. Responses are in black, with changes to the text in italics.

**Responses to Reviewer #1:**

The manuscript by Yu et al. is a valuable contribution to a long-term discussion on the role of spatial resolution in determining the total error of an atmospheric chemistry transport model. In brief, the authors focus on the analysis of an aircraft campaign in the South-Eastern US, which is a region characterized by large biogenic isoprene emissions, and compare the observations with simulations carried out with the same model (GEOS-Chem) at three spatial resolution (increasing from c.a. 400 to c.a. 28 km). They found a little impact by increased resolution on simulated values of ozone and NOx, especially in the free troposphere. They thus conclude that a coarse resolution model is adequate to resolve regional contributions to ozone export on a global perspective.

I found the manuscript concise, clear and well written, and I tend to agree with the main conclusions of the authors, based on the material showed here. I only have a few comments, which are more requests of clarification from an interested reader, that may further improve the interpretation of some of the results:

*Specific comments*

page 2, lines 19-29: in this paragraph the authors review previous similar studies on the horizontal resulution issue, but at the end of the manuscript it is not clear to the reader what is the advancement/difference/similarity (if any) with these studies. I recommend including a short paragraph in the conclusions on that.

We have updated the abstract, body of text, and conclusions to reflect these concerns.

In the abstract, the following text has been added:
*The good agreement of simulated and observed concentration variances implies that smaller-scale non-linearities (urban and power plant plumes) are not important on the regional scale.*

In the body of the text, on line 14 of page 6, we add the following:
*Non-linear effects have been reported in previous model studies at the km-scale of urban areas (Kumar and Russell, 1996; Chock et al., 2002; Shrestha et al., 2009; Tie et al., 2010) or pollution plumes (Henderson et al., 2010; Valin et al., 2011), and these small-scale effects are not resolved here even at $0.25^{o} \times 0.3125^{o}$ resolution.*

In the conclusions, we update the following paragraph (addition in italics):

Spatial correlations between model and observations improve as resolution increases from $4° \times 5°$ to $2° \times 2.5°$ but then decrease as resolution increases further to $0.25° \times 0.3125°$. This shows

that finer modes of variability are more difficult to capture in models of commensurate resolution, *consistent with the results of previous studies such as Fiore et al. (2003) and Valari and Menut (2008).*

And add the following sentence:
*Previous model studies have pointed out significant chemical non-linearities at the scale of urban and industrial plumes, smaller than the $0.25^o \times 0.3125^o$ resolution used here. However, the ability of GEOS-Chem to simulate the variability of concentrations observed in SEAC4RS implies that such small-scale effects are not important on the regional scale.*

page 5, lines 4-5: the varying ISOP-NOx relationship in the model is one of the most intriguing results presented in the manuscript. However, its interpretation is left too much to speculations by the reader. Can this analysis be improved? For example, can the referred statement on "temperature and stagnation" be proved calculating correlatons with temperature and a stagnation index?

We now compute correlations with temperature and update line 5 on page 5 to read as follows:
The 4◦ × 5◦ simulation actually shows positive correlations, likely reflecting a common sensitivity *to regional stagnation and resulting high temperatures. We find in the model that $NO_x$ concentrations are strongly correlated with surface air temperature on the $4^o \times 5^o$ scale (r = 0.66) but not on the $0.25^o \times 0.3125^o$ scale (r = 0.17).*

page 5, lines 13-15: The role of OH is recurring here and throughout the manuscript. Please consider to include some visualization of the changing OH or related species fields, this may help the reader in the interpretation of results.

We have updated Figure 2 to include panels colored by OH to show more clearly OH depletion under low-$NO_x$ conditions.

page 6, lines 26-27: I do not completely agree with this statement, in particular for isoprene. Looking at Figure 4, all the model realizations look pretty similar to observations, however Taylor diagram in Figure 5 display a dramatic decrease of model skills with increasing resolution. It is not clear to the reader why it happens at this point, it comes as some sort of surprise, so it needs further analysis and discussion. Perhaps, here and for other purposes it would be beneficial to include some sort of more direct visualization of the model-to-obs comparisons (e.g. simple timeseries of data, or scatter plots), maybe in the supplement, in order to keep the main text concise

We amended the text to clarify that correlations are computed using observations averaged to model grid and timestep. We average observations to model grid and timestep as this is a fairer way to compare model output to observations when computing correlations, but it also means correlations will worsen at higher resolution because higher resolution features are inherently more difficult to capture.
* * *
**Responses to Reviewer #2:**

The authors explain and show concisely the main thesis of the manuscript which falls within the scope of the journal. References and discussion of previous work is suffi- cient. I recommend publication with minor revisions.

*Main comments*

Many aspects (mostly chemical) are left to other papers which are not all available in ACPD yet (e.g. Travis et al.). Therefore, I would like to see in the manuscript the relative differences for the simulated daytime ozone levels going from 2∘x2.5∘ to 0.25∘x0.3125∘ resolution. Such a figure for August at the surface and at 4 km would be sufficient and help make the point of the authors clearer.

Travis et al., (2016) and the other GEOS-Chem papers referenced are now either published or in ACPD. References have been updated in our manuscript.

The authors discuss the segregation of isoprene and NOx emissions and of isoprene oxidation pathways. These statements need some abstraction by the reader and are obvious. They should be substantiated by calculations. The literature I am aware of discusses the intensity of chemical segregation between two species like isoprene and OH and not between two pathways or emissions. Please add mathematical definitions, calculations and a figure for different model resolutions.

Additional discussion of isoprene oxidation pathways can be found in Travis et al., (2016).

We have updated Figure 2 to include panels colored by OH to show more clearly OH depletion under low-NO$_x$ conditions (see response to reviewer 1).

**References:**

[revised manuscript text omitted]